# Genome and transcriptome evolve separately in recently hybridized *Trichosporon* fungi

Sira Sriswasdi [1,2,3], Masako Takashima[4,8], Ri-ichiroh Manabe[5], Moriya Ohkuma[4] & Wataru Iwasaki [1,6,7]

Genome hybridization is an important evolutionary event that gives rise to species with novel capabilities. However, the merging of distinct genomes also brings together incompatible regulatory networks that must be resolved during the course of evolution. Understanding of the early stages of post-hybridization evolution is particularly important because changes in these stages have long-term evolutionary consequences. Here, via comparative transcriptomic analyses of two closely related, recently hybridized *Trichosporon* fungi, *T. coremiiforme* and *T. ovoides*, and three extant relatives, we show that early post-hybridization evolutionary processes occur separately at the gene sequence and gene expression levels but together contribute to the stabilization of hybrid genome and transcriptome. Our findings also highlight lineage-specific consequences of genome hybridization, revealing that the transcriptional regulatory dynamics in these hybrids responded completely differently to gene loss events: one involving both subgenomes and another that is strictly subgenome-specific.

[1] Department of Biological Sciences, Graduate School of Science, the University of Tokyo, 2-11-16 Yayoi, Bunkyo-ku, Tokyo 113-0032, Japan. [2] Research Affairs, Faculty of Medicine, Chulalongkorn University, 1873 Rama 4 Road, Pathum Wan, Bangkok 10330, Thailand. [3] Computational Molecular Biology Group, Faculty of Medicine, Chulalongkorn University, 1873 Rama 4 Road, Pathum Wan, Bangkok 10330, Thailand. [4] Japan Collection of Microorganisms, RIKEN BioResource Research Center, 3-1-1, Koyadai, Tsukuba-shi, Ibaraki 305-0074, Japan. [5] Laboratory for Comprehensive Genomic Analysis, RIKEN Center for Integrative Medical Sciences, 1-7-22 Suehiro-cho, Tsurumi-ku, Yokohama, Kanagawa 230-0045, Japan. [6] Department of Computational Biology and Medical Sciences, Graduate School of Frontier Sciences, the University of Tokyo, 5-1-5 Kashiwanoha, Kashiwa-shi, Chiba 277-8568, Japan. [7] Atmosphere and Ocean Research Institute, the University of Tokyo, 5-1-5 Kashiwanoha, Kashiwa-shi, Chiba 277-8564, Japan. [8] Present address: Department of Microbiology, Meiji Pharmaceutical University, Kiyose, Tokyo 204-8588, Japan. Correspondence and requests for materials should be addressed to S.S. (email: sira.sr@chula.ac.th) or to W.I. (email: iwasaki@bs.s.u-tokyo.ac.jp)

Allopolyploidy, or genome hybridization, is an evolutionary event that involves merging of two or more distinct genomes into the same organism. Such expansion of gene repertoire relaxes evolutionary constraints on homeologous genes—or groups of gene copies derived from different parents—in the descendent species, and facilitates the emergence of new gene functions and expression regulations[1–3]. As a result, genome hybridization is an important force that gives rise to species with novel phenotypes and capabilities, which have become essential in agriculture, food industry, and biotechnology. Nonetheless, merging distinct genomes also brings together genes with incompatible regulatory networks and protein products[4,5]. This phenomenon, often called genome shock or transcriptome shock, induces complex reprogramming of gene expression that distinguishes inter-species genome hybridization from other, less disruptive types of polyploidization processes[2,6,7].

To date, the evolutionary mechanisms that shaped homeolog expression in a wide range of naturally occurring and synthetic eukaryotic hybrids, including plants[8–12], fish[13], and fungi[14–16], have been characterized. These studies also revealed that post-hybridization transcriptional regulation of homeolog expression varies markedly across different hybrid species. While some hybrids exhibited strong genome-wide or tissue-specific biases toward particular homeologs or parental genomes[6,17], others underwent rather conservative evolution with reduction in expression divergence among homeologs[15]. A key question is to what extent are transcriptional expression reprogramming driven by factors such as the degree of parental divergence and the timing of hybridization event—in other words, how history repeats itself when it comes to the evolution of inter-species hybrids. Although some striking similarities in homeolog expression pattern were found between eukaryotic hybrids as distant as fungi and plants[15], others have shown that genetic background contributed heavily to the evolutionary outcomes in individual lineages[10,11,18]. The ability to distinguish between universal and lineage-specific consequences of post-hybridization genome evolution is thus essential for gaining further insights into this important evolutionary phenomenon.

Recently, we discovered two recent and independent genome hybridization events in the genus *Trichosporon* of Basidiomycota fungi, and sequenced the genomes of the two diploid, asexually reproducing hybrids, *Trichosporon coremiiforme* and *T. ovoides*, and their close relatives[19,20]. This revealed that *T. coremiiforme* descended from two closely related parental species with 7% amino acid sequence divergence, while *T. ovoides* descended from more distant parental species with 17% divergence. Both hybrids retain more than 70% of homeolog pairs and are likely still in the early stage of post-hybridization evolution. Our prior study also suggested that the difference in parental divergence was enough to induce subgenomic dominance in *T. ovoides*, resulting in twice as many gene losses from one of its subgenome compared to the other, but not in *T. coremiiforme*. Therefore, these species constitute a key platform for investigating not only the mechanisms responsible for genome stabilization but also the reproducibility of such processes in closely related lineages.

In this study, we performed RNA sequencing to compare the transcriptome profiles and characterized general patterns in homeolog expression levels and evolutionary rates. On one hand, we consistently observed increased sequence conservation and low expression divergence after genome hybridization among evolutionarily conserved and highly expressed duplicated homeologs, as well as a lack of concerted evolution at sequence level and expression level. On the other hand, opposite transcriptional stoichiometry preservation mechanisms in the two hybrids are also revealed. Our findings illustrate that genome and transcriptome stabilizations are distinct evolutionary processes in

young polyploids and that closely related hybrids may follow similar evolutionary paths in some respects but at the same time adopt completely different mechanisms in the others.

## Results

**Transcriptome profiling of hybrid *Trichosporon* fungi.** Using RNA sequencing, we were able to measure the expression levels of more than 94% of the predicted genes in five *Trichosporon* species—the hybrid *T. coremiiforme* and *T. ovoides*, and the non-hybrid *T. asahii*, *T. faecale*, and *T. inkin*—under log-phase and stationary-phase growth conditions (see the "Materials and methods" section). The reproducibility is high across replicates and transcripts belonging to homeologs from different sub-genomes of a hybrid species could be distinguished (Supplementary Fig. 1). The assignment of homeologs to subgenomes A and B in *T. coremiiforme* and *T. ovoides* were performed using a combination of conserved gene order structure, phylogenetic reconstruction, and amino acid sequence similarity as detailed in our prior study (see ref. [19] and also section "Materials and methods"). Overall, 5130 and 5083 genes were assigned to sub-genomes A and B of *T. coremiiforme*, and 4862 and 4386 genes were assigned to subgenomes A and B of *T. ovoides*, respectively. *T. coremiiforme*'s subgenomes contain similar number of single-copy genes (444 and 397 genes, respectively), while subgenome A of *T. ovoides* contains almost two times the number of single-copy genes than subgenome B does (959 and 483 genes, respectively). Then, we evaluated the correlations of transcript expressions between homeologs of each hybrid species and their orthologs in non-hybrid relatives. In concordance with previous findings based on genomic data[19,20], transcript expression data here also show that *T. asahii* and *T. inkin* are the closest relatives to both of *T. coremiiforme*'s subgenomes and both of *T. ovoides*'s sub-genomes, respectively (Fig. 1 and Supplementary Fig. 2). Therefore, we selected *T. asahii* as the reference non-hybrid for *T. coremiiforme* and selected *T. inkin* for *T. ovoides*.

**Gene expression convergence between hybrid subgenomes.** The extent of gene expression correlation across genomes and sub-genomes closely follows phylogenetic relationship between *Trichosporon* species, with higher correlation between more closely related genomes and subgenomes (Fig. 1 and Supplementary Fig. 2). We also observed significantly higher expression correlations between subgenomes within a hybrid species compared to those across closely related species (Fig. 1a–d). This trend is particularly striking for *T. ovoides*, whose subgenome A is evolutionarily closer to *T. inkin*'s genome than to its subgenome B counterpart (10.0% and 16.7% median amino acid sequence divergence, respectively). Expression levels of homeologs from subgenome A of *T. ovoides* are more correlated with expression levels of *T. inkin*'s orthologs than those of subgenome B do, as expected. However, the two subgenomes of *T. ovoides* exhibit much stronger correlation with each other (Fig. 1d and Supplementary Fig. 2). Given the degree of evolutionary divergence between parental species of *T. ovoides* (Fig. 1a), which resulted in high gene loss rates predominantly from one of its subgenomes[19], the high correlation of inter-subgenome transcript expression suggests that either transcriptional regulations on the two sub-genomes converged rapidly or the rate of homeolog expression divergence slowed down after genome hybridization event.

We further investigated the mechanisms behind highly correlated transcriptional activity across hybrid subgenomes by characterizing the presence of transcription factor genes and their corresponding binding sites in *Trichosporon*. Using the well-annotated *Saccharomyces cerevisiae* as reference, we were able to identify orthologs of 18 known transcription factors, namely

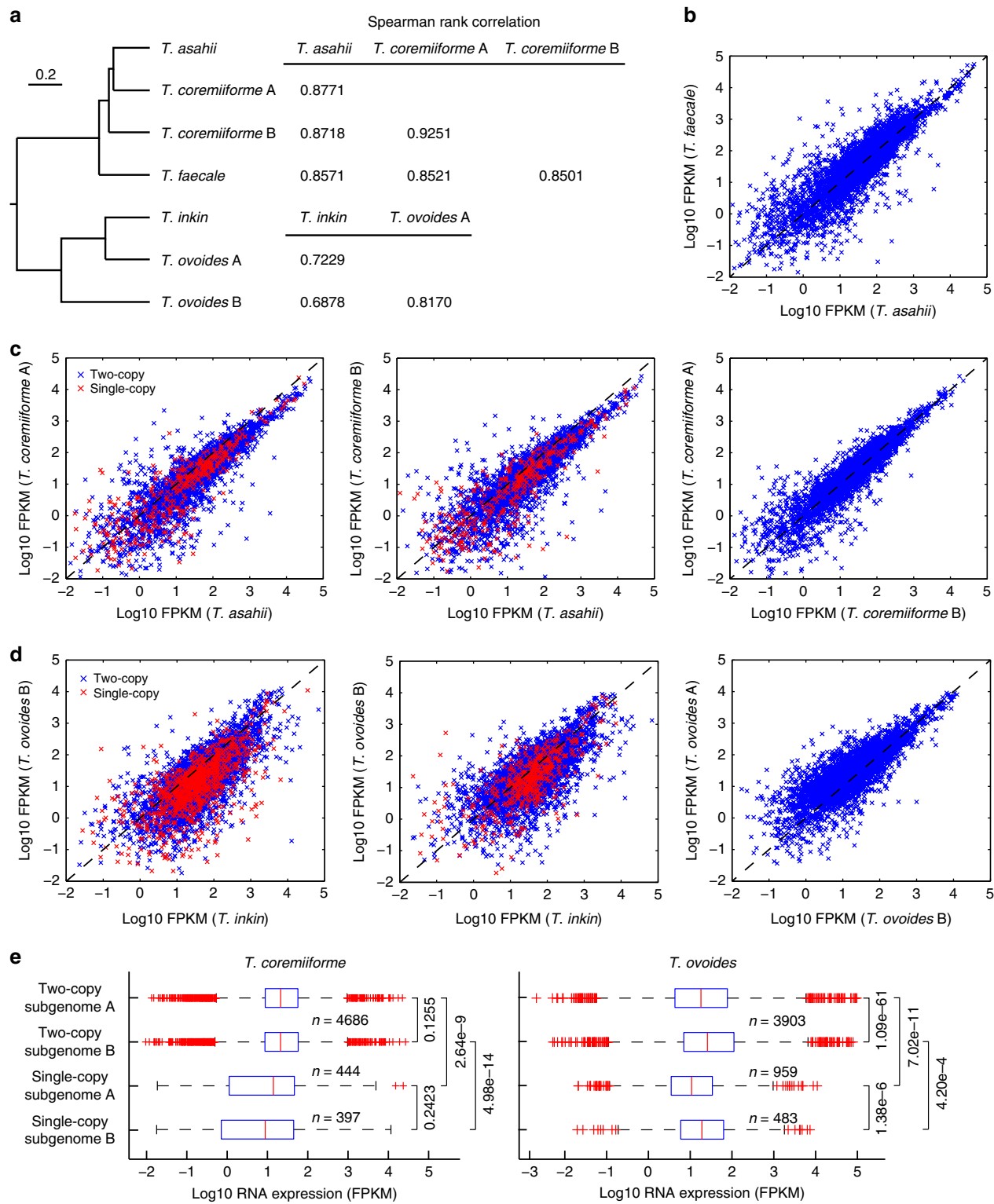

ARG81, ARO80, ASG1, CHA4, GAL4, HAP3, KAR4, MBP1, NHP6A/NHP6B, PIP2, PPR1, PUT3, RDS2, SKN7, SPT15, STB5, TEA1, and UGA3, in *Trichosporon* species (Supplementary Table 1). The majority of these transcription factors remain duplicated in *T. coremiiforme* while 8 out of 18 have become single-copy in *T. ovoides*. Then, we searched for binding sites of these transcription factors in the 1 kb upstream regions from the predicted start codon of all two-copy homeolog pairs in both hybrids (see the "Materials and methods" section). This revealed

significant sharing of common transcription factor-binding sites between homeologous genes that likely contributed to their concerted transcriptional activities (Supplementary Table 2, hypergeometric test *p*-value < 1.05e−3 for all transcription factors). Similar enrichments were observed within 300-bp upstream regions.

Next, to look for signs of subgenomic dominance which often manifest in the form of large-scale transcriptional activity bias toward homeologs from specific subgenomes, we compared

**Fig. 1** Rapid convergences of transcriptional regulation of homeologous genes following genome hybridization. Expression levels from log-phase growth condition are shown. **a** Phylogenetic relationship between *Trichosporon* species analyzed. Spearman rank correlation coefficients for pairwise comparisons of expression levels between orthologs and homeologs from different *Trichosporon* genome and subgenomes are indicated. Only gene ortholog groups that are present in all species involved (*T. asahii*, *T. faecale*, and *T. coremiiforme* for the top table and *T. inkin* and *T. ovoides* for the bottom table) were included in the calculations. **b** Scatter plot comparing expression levels in log10 fragments per kilobase of transcript per million mapped reads (FPKM) between *T. asahii* and *T. faecale* orthologs. Dashed lines indicate the $x = y$ diagonal. **c** Similar scatter plots for the comparisons of expression levels between *T. coremiiforme*'s subgenomes and *T. asahii*. Data points corresponding to two-copy and single-copy homeolog groups in *T. coremiiforme* are distinguished by blue and red markers, respectively. **d** Similar scatter plots for the comparisons of expression levels between *T. ovoides*'s subgenomes and *T. inkin*. **e** Boxplots comparing subgenome-specific expression levels in *T. coremiiforme* and *T. ovoides*. Wilcoxon signed-rank test *p*-values for the paired comparisons of expression levels among two-copy homeologs and Mann–Whitney *U*-test *p*-values for the comparisons among single-copy genes are indicated. Blue boxes indicate the 25th–75th percentile ranges. Red bars indicate the medians. Black whiskers indicate the approximated 0.35th–99.65th percentile ranges. Red cross markers indicate individual data points lying outside the 0.35th–99.65th percentile ranges. Number of genes in each group is indicated

homeolog expression levels between subgenomes A and B in *T. coremiiforme* and *T. ovoides* using paired tests for two-copy homeolog pairs and unpaired tests for single-copy genes. Homeologs from the two subgenomes of *T. coremiiforme* exhibit similar expression levels (Fig. 1e), in good agreement with prior observation of balanced gene loss from the two subgenomes[19]. In contrast, even though transcriptional activities in *T. ovoides* are significantly higher on subgenome B (Fig. 1e), many more gene losses (959 out of 1442 genes lost) including ribosomal protein coding genes (10 out of 13 genes lost)[19] and transcription factors (7 out of 8 genes lost, Supplementary Table 1) also occurred on this subgenome. In light of these conflicting evidences of subgenomic dominance in *T. ovoides*, because gene expression may be further moderated at many stages beyond transcription[14] whereas the effect of gene deletion is permanent, we believe that subgenome A, which had lost much fewer genes, is the dominant subgenome of *T. ovoides* and that the higher transcriptional activity of subgenome B is rather due to stronger inherited cis-regulatory elements from its parent. Gene functional enrichment analyses of homeolog groups that are more transcriptionally active on subgenome B did not reveal any significant term (see the "Materials and methods" section).

There were small numbers of two-copy homeolog pairs (32 in *T. coremiiforme* and 25 in *T. ovoides*), where both gene copies remain in their respective subgenomes, but one gene copy appeared to be transcriptionally silent (i.e., could not be detected via RNA sequencing). However, this was likely due to the detection limit because the remaining gene copies in these homeolog groups tend to have very low-expression levels (<1 FPKM). We also identified a 370 kb region in *T. coremiiforme*'s scaffold 7, where transcriptional activities are distinctively suppressed (Supplementary Fig. 3). However, functional enrichment analysis of 138 two-copy homeolog pairs located in this region did not reveal any significant enrichment.

**Lack of concerted evolution of gene sequence and expression.** To investigate the interaction between sequence evolution and transcriptional evolution, we calculated the divergence in sequence evolutionary rate, defined as the ratio of nonsynonymous substitution rate (d*N*) to synonymous substitution rate (d*S*), or d*N*/d*S*, and divergence in expression level for each homeologous gene pair. Divergences were calculated as the ratio of subgenome A over subgenome B and normalized to remove intrinsic biases that might be inherited from parental species or those that might result from unequal gene loss's effects on the evolution and expression of remaining genes. Furthermore, we employed a statistical test to control for large divergences in evolutionary rate that might arise from extremely small values of d*N* or d*S* (see the "Materials and methods" section), which revealed that many of the observed large divergences in evolutionary rate are in fact not statistically significant (Fig. 2a, d, black

data points with high absolute d*N*/d*S* divergences). Overall, there was neither significant correlation between the two divergence measures in either species nor significant overlap between the sets of homeologs with divergent evolutionary rate and those with divergent expression level (Fig. 2a, c, Supplementary Fig. 4, Supplementary Data 1 and 2, Spearman rank correlations between the two divergence measures range from −0.1220 to 0.0241, hypergeometric test *p*-values for the overlap range from 0.0041 to 0.3978). The lack of concerted evolution of gene sequence and expression, such as reduction in evolutionary rate of the homeolog copy with higher expression level, supports the hypothesis that these *Trichosporon* hybrids are still in the early stage of post-hybridization evolution.

Our prior study has shown that the deceleration of evolutionary rate (defined as the situation where d*N*/d*S* ratios of homeologs in a hybrid species are significantly lower than d*N*/d*S* ratio of their ortholog in the non-hybrid reference) is widespread in *Trichosporon* hybrids and is likely part of evolutionary mechanisms to preserve gene integrity and stabilize hybrid genome[19]. Here, we found that deceleration of evolutionary rate also occurred on homeolog pairs with significantly higher expression levels compared to others, especially in *T. coremiiforme* (Fig. 2b, d and Supplementary Fig. 4). In comparison, homeolog pairs with divergent evolutionary rates are not strongly enriched for genes with high or low expression levels. Divergence of expression level, on the other hand, occurred on homeolog pairs with significantly lower expression levels and higher mutation and evolutionary rates compared to others (Supplementary Fig. 5). Interestingly, the sets of homeolog pairs with divergent expression levels in the two hybrid species significantly overlap and are enriched for transmembrane transporters (Supplementary Table 3).

**Two modes of stoichiometric maintenance driven by divergence.** A major evolutionary response in hybrid genomes is the reestablishment of gene expression stoichiometry among homeologs that were differently regulated in the parental species and homeologs whose protein products are incompatible with each other. To investigate this phenomenon, we inferred protein–protein interactions between *Trichosporon* genes using *S. cerevisiae*'s interactome as reference (see the "Materials and methods" section). Overall, 9957 interactions could be mapped to *T. coremiiforme* and 10,969 interactions could be mapped to *T. ovoides*. Out of these interactions, 2181 and 3109 occur between homeolog groups with different gene copy numbers (i.e., interaction involving a single-copy gene and a two-copy homeolog pair) in *T. coremiiforme* and *T. ovoides*, respectively (Fig. 3a, Supplementary Data 3 and 4). To measure the extent of stoichiometry preservation across *Trichosporon* species, transcript expression stoichiometry between homeolog groups that form protein–protein interaction partners in each hybrid species were

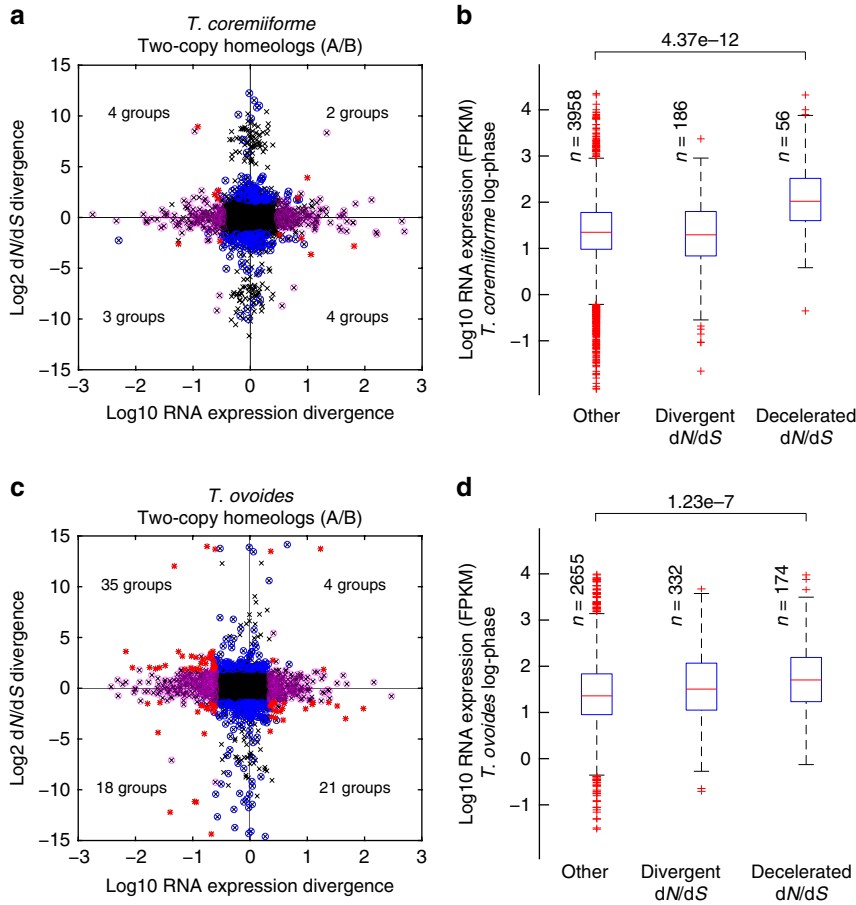

**Fig. 2** Lack of concerted evolution at sequence and expression level among two-copy homeolog groups. **a** Scatter plot comparing divergence in evolutionary rate (dN/dS ratio) and divergence in expression level for two-copy homeolog pairs in *T. coremiiforme*. Divergences were calculated as the ratios of subgenome A homeolog's over subgenome B homeolog's. Black x markers display the data for all two-copy homeolog pairs. Blue and magenta circles indicate homeolog pairs with significant divergence in only evolutionary rate or only expression level, respectively (adjusted *p*-value ≤ 0.01 and fold-difference ≥ 3, see the "Materials and methods" section). Red asterisks indicate homeolog groups with significant divergence in both evolutionary rate and expression level and the number of these homeolog groups are indicated in each quadrant. Expression levels from log-phase growth condition are shown. **b** Box plots showing log-phase expression level of two-copy homeolog pairs in *T. coremiiforme* with divergent evolutionary rates or decelerated evolutionary rates compared to *T. asahii*'s orthologs (see the "Materials and methods" section). Mann–Whitney *U*-test *p*-value for the comparison between homeolog pairs with decelerated evolutionary rates and those without is indicated at the top. The numbers of homeolog pairs belonging to each class are indicated next to the corresponding box plot. Blue boxes designate the 25th–75th percentile ranges. Red bars indicate the medians. Black whiskers designate the approximated 0.35th–99.65th percentile ranges. Red cross markers indicate individual data points lying outside the 0.35th–99.65th percentile ranges. **c** and **d** Similar plots for *T. ovoides*–*T. inkin* comparison

compared to the corresponding transcript expression stoichiometry between orthologs in non-hybrid relatives (namely, *T. asahii* for *T. coremiiforme* and *T. inkin* for *T. ovoides*). For each protein–protein interaction, we further calculated the ratio between transcript stoichiometry in a hybrid species and the stoichiometry in the non-hybrid reference (stoichiometry ratio in Fig. 3b–e). This revealed that the transcriptional stoichiometry between protein–protein interaction partners are mostly conserved (Fig. 3b, d). The high variance of *T. coremiiforme*-to-*T. asahii* stoichiometry ratio for protein–protein interactions involving two single-copy genes is likely due to low number of such cases (Fig. 3a, b).

Unexpectedly, for transcript stoichiometry of protein–protein interactions that involve homeolog groups with different gene copy numbers (i.e., those involving one single-copy gene and one two-copy homeolog pair), the two hybrids exhibited different patterns. In *T. coremiiforme*, transcript expression level of the single-copy interaction partner is in stoichiometric balance (compared to stoichiometry in *T. asahii*) with the total expression

level of the two-copy interaction partner instead of being in stoichiometric balance with the expression level of just one gene copy of the two-copy interaction partner that is located on the same subgenome (Fig. 3c). On the other hand, such conservation of transcriptional stoichiometry based on total expression levels of homeologs from both subgenomes is absent in *T. ovoides* (Fig. 3e). Here, the expression level of the single-copy interaction partner in *T. ovoides* is in stoichiometric balance (compared to stoichiometry in *T. inkin*) with the expression level of only one gene copy of its interaction partners that is located on the same subgenome. A possible explanation is that conservation of transcriptional stoichiometry is driven by the degree of compatibility between the single-copy gene and the two copies of the homeolog pair it forms protein–protein interaction with. In other words, if the protein coded by the single-copy gene can interact with only one of the two species of proteins coded by its two-copy interaction partner, then the interaction stoichiometry would not involve the expression level of the incompatible gene copy. However, we did not observe any difference in stoichiometry

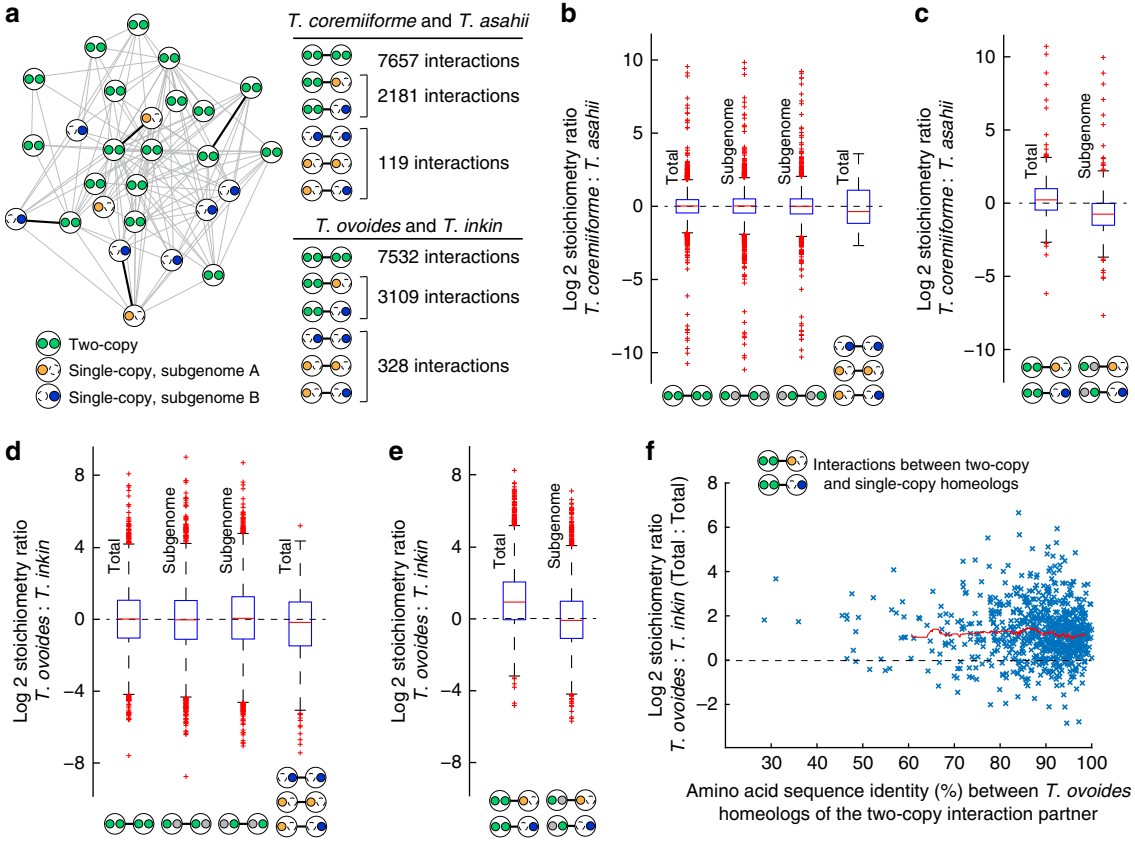

**Fig. 3** Two modes of transcriptional stoichiometry maintenance following genome hybridization. Expression levels from log-phase growth condition are shown. **a** Schematic of a protein–protein interaction network in hybrid species. Large circle represents a homeolog group with inner circles representing subgenome-specific homeologs. Dashed outline for inner circle indicates gene loss and solid color indicates presence. Green is used when both homeologs are present. Orange or blue is used when only subgenome A or subgenome B homeolog is present, respectively. The numbers of interactions in each category (two-copy to two-copy, two-copy to single-copy, and single-copy to single-copy) are listed. **b** Box plots comparing the conservation of interaction stoichiometry across *T. coremiiforme* and *T. asahii*. From left to right, the data for (i) interactions between two-copy homeologs, (ii) interactions between two-copy homeologs but considering only the transcript level of subgenome A homeologs, (iii) same as (ii) but considering only the transcript level of subgenome B homeologs, and (iv) interactions between single-copy homeologs are shown. Symbols at the bottom indicate the different sets of interactions considered in each box plot. Gray circles indicate homeologs that are present but were excluded from stoichiometry calculation in order to highlight subgenome specificity. Expression levels from log-phase growth condition are shown. Vertical labels indicate the type of stoichiometry under consideration (Total = stoichiometry involving all homeolog copies, Subgenome = stoichiometry involving only homeolog copies belonging to the same subgenome). Blue boxes designate the 25th–75th percentile ranges. Red bars indicate the medians. Black whiskers designate the approximated 0.35th–99.65th percentile ranges. Red cross markers indicate individual data points lying outside the 0.35th–99.65th percentile ranges. **c** Boxplots comparing the conservation of stoichiometry across *T. coremiiforme* and *T. asahii* for interactions between a two-copy homeolog group and a single-copy homeolog. **d** and **e** Similar box plots for *T. ovoides*–*T. inkin* comparisons. **f** Scatter plot showing the relationship between the conservation of stoichiometry across *T. ovoides* and *T. inkin* for protein–protein interactions involving a two-copy homeolog pair and a single-copy gene and the amino acid sequence identity between the homeolog copies within the two-copy group. Red trend line indicates the running median

conservation among interactions involving homeolog pairs with various degrees of sequence conservation (Fig. 3f).

In the context of protein–protein interaction network, gene losses in *T. coremiiforme* and *T. ovoides* also displayed different preferences. Among interactions between a single-copy gene and a two-copy homeolog pair in *T. coremiiforme*, the single-copy gene tended to be more highly expressed (Supplementary Fig. 6 and Supplementary Data 5, 1522 out of 2181 such interactions). For this comparison, the expression levels of orthologs of *T. coremiiforme*'s genes in *T. asahii* was used instead of the expression levels in *T. coremiiforme* in order to avoid evolutionary impacts of genome hybridization on gene expression. In *T. ovoides*, we found that protein–protein interaction subnetworks surrounding single-copy genes on subgenome A were significantly more densely connected than subnetworks surrounding single-copy genes on subgenome B (Supplementary Fig. 6, Mann–Whitney *U*-test *p*-value = 0.0057, 1.32 folds difference in median clustering coefficient).

## Discussion

Here, we characterized evolutionary consequences of genome hybridizations on gene transcript expression in two closely related natural hybrids, *T. coremiiforme* and *T. ovoides*, of Basidiomycota fungi. Comparative transcriptome profiling via RNA sequencing revealed shared conservative patterns in the transcript expression, reinforcing the notion that genome stabilization is a key evolutionary force in recently hybridized species. High correlation of homeolog transcript expression (Fig. 1c, d), especially that across *T. ovoides*' evolutionarily distant subgenomes, indicates swift reconciliation of parental transcriptional regulatory networks[15]. Deceleration of evolutionary rates, which would protect the sequence and functional integrity of genes, was found to act on highly expressed homeologs (Fig. 2b, e), in good agreement with prior studies of polyploidization[21–23]. Although significant enrichment of transmembrane transporters among homeolog pairs with divergent transcript expression (Supplementary

Table 3) may reflect an adaptation mechanism against shifts in surface-to-volume equilibrium due to the enlarged genome[24], these transporters did not seem to correspond to specific molecule or ion types that could indicate the underlying mechanisms (Supplementary Table 4).

The conservation of transcriptional stoichiometry between a two-copy and a single-copy protein–protein interaction partners in *T. coremiiforme* agrees well with the dosage subfunctionalization model[25], which proposed that once the expression level of one homeolog became high enough, the lower expressed homeolog could be lost with minimal selective pressure, as well as prior observations in other polyploids that achieving dosage balance is a major evolutionary concern[26,27]. It is also notable that in the context of *T. coremiiforme*'s protein–protein interaction network, homeolog groups that have lost a gene tended to be more highly expressed than their interaction partners (Supplementary Fig. 6). This may be because a reduction in molar concentration of protein interaction partner that are more highly expressed results in less total amount of unbounded proteins than the same percentage reduction in concentration of the interaction partner with lower expression[28,29] (Supplementary Fig. 6). Even though protein–protein interaction stoichiometry is ultimately regulated at protein level and changes at transcript level can be offset by changes in translational regulation[14], our results show evidence of strong regulatory responses to stoichiometric alteration at transcript level[30,31].

Characterization of *T. ovoides* genome has shown that subgenome A lost significantly less genes than subgenome B did and therefore is likely to be the dominant subgenome[19]. This discrepancy in the amount gene losses may also underlie the systematic differences in evolution and expression between the two subgenomes of *T. ovoides* that were illustrated in this study. Preferential retention of homeologs from subgenome A in dense regions of the protein–protein interaction network (Supplementary Fig. 6) also supports this conclusion. However, contrary to prior findings in other hybrids that the dominant subgenomes would also be more highly expressed[32,33], subgenome A of *T. ovoides* has significantly lower expression level than subgenome B does (Fig. 1e). Furthermore, while *T. ovoides* exhibits considerable degree of global gene expression convergence (Fig. 1d), the subgenome-specific pattern of local protein–protein interaction stoichiometry clearly highlights incompatibility between the two subgenomes (Fig. 3e). Our observation that the sequence similarity between subgenome A and subgenome B homeolog copies has no impact on stoichiometry conservation (Fig. 3f) suggests that the causes of these incompatibilities are likely not at the protein functional level but rather at regulatory level. More details on gene regulatory networks, epigenetics, and protein functions would be required to unravel the mystery of post-hybridization evolution in *Trichosporon* hybrids beyond sequence-level and dosage-level constraints[11,15,27].

## Materials and methods

**Genome sequencing and annotation**. Sequencing of genomic DNA of *T. asahii* JCM 2466, *T. coremiiforme* JCM 2938, *T. ovoides* JCM 9940, *T. faecale* JCM 2941, and *T. inkin* JCM 9195 strains was previously performed[19]. Raw reads and assembled genome sequences are available at GenBank/EMBL/DDBJ under accession PRJDB3696 for *T. asahii*, PRJDB3698 for *T. faecale*, PRJDB3697 for *T. coremiiforme*, PRJDB3701 for *T. inkin*, and PRJDB3702 for *T. ovoides*. Protein-coding genes were also previously predicted using GeneMark-ES version 2[34]. Briefly, the hidden Markov model for GeneMark-ES was trained using previously published *T. asahii* CBS 2479's genome sequence[35] and using default parameters. Genes that translate to <100 amino acids in length were discarded. The numbers of predicted genes for non-hybrid species ranged from 6733 in *T. inkin* to 7797 in *T. asahii* and 7804 in *T. faecale*. The number of predicted genes for hybrid species are 12,877 for *T. ovoides* and 13,398 for *T. coremiiforme*.

**Ortholog group and subgenome assignments**. Orthologous gene clustering and subgenome assignment for *Trichosporon* genomes were performed exactly as previously described[19], with the only difference being that our own draft genome sequence for *T. asahii* strain JCM 2466 was used instead of the sequence for strain CBS 2479. Gene ortholog relationships across *Trichosporon* genomes were determined using inParanoid version 4.1[36] and MultiParanoid[37]. Ortholog groups that contained more than one gene in a non-hybrid genome (*T. asahii*, *T. faecale*, and *T. inkin*) or more than two genes in a hybrid genome (*T. coremiiforme* and *T. ovoides*) were removed from further analyses in order to prevent the complication of distinguishing between in-paralogs and out-paralogs. At this step, 7509 out of 7857 ortholog groups identified by MultiParanoid were retained.

Gene Order Browser[38] was then used to identify conserved syntenic regions across genomes. Genes in the hybrid genomes (*T. coremiiforme* or *T. ovoides*) were assigned to either subgenome A or B based on a combination of syntenic structures, sequence identities, and phylogenies exactly as previously described[19]. For *T. coremiiforme*, because its homeologs are almost always similar to their ortholog in *T. asahii* (median difference in sequence identity to *T. asahii* = 1.3% at nucleotide level), we assigned its genes to subgenomes according to the consensus assignment at the scaffold level. Synteny structures supported by at least 10 homeologous gene pairs were used to separate *T. coremiiforme* scaffolds into two subgenome tracks. RAxML version 8.2.11[39] was then used to determine the most likely phylogeny and place *T. coremiiforme* subgenome tracks as A or B according to their evolutionary distances from *T. asahii* in the resulting phylogenetic trees. The general time-reversible coupled with rate heterogeneity among sites (-m GTRGAMMA) model was used and bootstrap count was set at 1000 (-# 1000). For *T. ovoides*, the difference in evolutionary distances between its two parental genomes from *T. inkin* is large enough that each gene in a homeologous pair could be assigned to subgenome A or B based on the difference in sequence identity levels to their common ortholog in *T. inkin*. Single-copy genes in *T. ovoides* were then assigned to subgenomes based on the consensus assignment of 20 nearby homeologous gene pairs. It should be noted that genes that do not belong to a synteny structure and are not located near other genes could not be assigned to a subgenome and were removed from further analyses (1161 genes from *T. coremiiforme* and 735 genes from *T. ovoides*).

**RNA sequencing and alignment**. RNA samples were extracted from each species under both the log-phase and the stationary-phase growth conditions. Briefly, cells grown in YM broth medium (BD-Difco) at 30 °C were harvested after 17–18 h (OD = 0.7–1.0, except *T. ovoides*) and 90–91 h incubation for log-phase and stationary-phase samples, respectively. The optical density at 660 nm were monitored with the shaking incubator (TVS 126MA; Advantec Toyo). Samples of *T. ovoides* were harvested at the same time with other samples, since their optical density could not be determined due to cell aggregation. RNA was extracted using a combination of Sepasol reagent (Nacalai Tesque, Inc., Kyoto, Japan) with glass bead disruption and RNeasy kit (Qiagen, Hilden, Germany). Transcript libraries were prepared from 1 μg total RNA using TruSeq Stranded mRNA Library Prep Kit (Illumina, San Diego, USA) according to the kit's protocol. Sequencings were performed on Illumina HiSeq 2500 on a high output run mode to generate 100-base paired-end reads. Sequencing yields range from 25M to 30M read pairs per sample, with quality scores (Illumina Q scores) consistently above 35.23. Two biological replicates were collected and subjected to sequencing in each case.

Paired-end reads were aligned to assembled genomes (see the section "Genome sequencing and annotation") using MapSplice version 2.1.8[40] and subsequently processed using Cufflinks version 2.2.1[41] with default parameters. We followed the protocol and specific Cufflinks commands as detailed in ref. 41. Kallisto version 0.43.0[42] was also used in parallel to Cufflinks to evaluate the reproducibility of detected transcripts. The bias correction option in Kallisto (–bias) was enabled. Transcript abundances from Cufflinks in fragments per kilobase of transcript per million mapped reads (FPKM) were log-transformed and averaged across replicates. Overall, we could detect transcripts corresponding to more than 94% of the predicted genes in each species. Furthermore, transcript abundances are highly consistent across replicate samples and across analysis software (Supplementary Fig. 1). When comparing transcript abundances between genes on different subgenomes, we found that genes that could not be assigned to a subgenome exhibit unusually high transcript abundances. This suggested that the copy numbers of these genes might be underestimated, possibly due to misassembly. Therefore, we decided to remove these genes from further consideration.

To identify homeolog pairs in hybrid species whose transcript expression significantly diverged, we used DESeq2 in R version 3.5.2[43] to process raw read count data. Wald test with adjusted *p*-value cutoff of 0.01 and a fold difference threshold of three-fold was applied. It should be noted that DESeq2 automatically normalizes data across samples as part of its pipeline.

**Evolutionary rate calculations**. Phylogenetic relationship between *Trichosporon* genomes and subgenomes (for the cases of hybrids) had been previously elucidated[19,20] (Fig. 1a). The codeml module of PAML version 4.9[44] was then used to estimate the synonymous and non-synonymous substitution rates (d$S$ and d$N$) for genes in each ortholog group according to this known phylogeny. The free-ratio model which allow d$N$/d$S$ to vary across branches was used (model = 1, NSsites = 0). Codon frequency model F3X4 was selected (CodonFreq = 2). The molecular

clock model was disabled (clock = 0). The phylogenetic tree estimated by RAxML from concatenated alignment of all ortholog groups was input as the initial tree. MUSCLE version 3.8.31[45] was used to align amino acid sequences and the resulting alignments were mapped to nucleotide sequences to create codon-level multiple sequence alignments. The dN/dS ratio along the phylogenetic branch directly leading to each gene was taken as that gene's evolutionary rate. To filter out ortholog groups with saturated substitutions, groups containing genes with estimated dS or dN of 2 or larger were removed from further consideration (32 groups from *T. asahii*–*T. faecale*–*T. coremiiforme* comparison and 560 groups from *T. inkin*–*T. ovoides* comparison). Homeologs with decelerated evolutionary rates in *T. coremiiforme* and *T. ovoides* were defined as those with at least three-fold lower evolutionary rates compared to their respective non-hybrid orthologs.

To identify homeologous groups in a hybrid species in which the evolutionary rates of subgenome A and subgenome B gene copies have significantly diverged, we calculate the *p*-value under the null hypothesis that both homeolog have the same evolutionary rate via the following approximation. For a pair of homeologs with $N$ nonsynonymous sites, $S$ synonymous sites, $dN_1$ and $dN_2$ observed nonsynonymous substitution rates, $dS_1$ and $dS_2$ observed synonymous substitution rates, and a common evolutionary rate $\omega$, we model the number of observed nonsynonymous substitutions $N\,dN_i$ as coming from a binomial distribution $B(N, dS_i\omega)$. Because $N$ are generally large, $B(N, dS_i\omega)$ can be approximated by a normal distribution with mean $NdS_i\omega$ and variance $NdS_i\omega(1 - dS_i\omega)$. Hence, the observed evolutionary rates $dN_i/dS_i$ follows the normal distribution with mean $\omega$ and variance $\frac{\omega(1-dS_i\omega)}{NdS_i}$. This means that the difference between observed evolutionary rates, $dN_1/dS_1 - dN_2/dS_2$, is approximately normally distributed with mean 0 and variance $\frac{\omega(1-dS_1\omega)}{NdS_1} + \frac{\omega(1-dS_2\omega)}{NdS_2}$. Finally, as the dN and dS were independently estimated for each homeolog group, we applied a Benjamini–Hochberg procedure on the *p*-values to control the false discovery rate at 1% for identifying significant divergence in evolution rates. To account for intrinsic difference in evolutionary rates between the two subgenomes of *T. ovoides* that might be inherited from the parental species, prior to performing the *p*-value calculation described above, we normalize the evolutionary rates on one subgenome by a constant factor so that the median of subgenome A-to-subgenome B evolutionary rate ratio is one. Finally, a three-fold threshold was also applied to select for homeolog groups with statistically significant evolutionary rate divergences that also exhibit at least three-fold difference in evolutionary rates between subgenome A and subgenome B gene copies.

**Protein–protein interaction and stoichiometry analyses**. To infer protein–protein interactions in *Trichosporon*, *T. asahii* and *T. inkin* genes were mapped to their orthologs in *S. cerevisiae* using inParanoid. Only gene ortholog groups that are present in all three species were retained. An *S. cerevisiae* protein–protein interaction dataset was downloaded from the Saccharomyces Genome Database (https://downloads.yeastgenome.org/curation/literature/interaction_data.tab)[46]. Self-loops and duplicated interactions were removed. Overall, 14,686 interactions between *Trichosporon* genes were inferred. *T. coremiiforme* contains 9957 of these interactions and *T. ovoides* contains 10,969 interactions. In hybrid *T. coremiiforme* and *T. ovoides* whose genes may exist as single-copy or as a part of homeologous pairs, each protein–protein interaction was further classified based on the status of its interaction partners (Fig. 3, ortholog group's statuses are two-copy, single-copy on subgenome A, or single-copy on subgenome B). The NetworkAnalyzer module of Cytoscape version 3.6.1[47] was used to calculate the clustering coefficient for each protein on the original *S. cerevisiae* interaction network.

The stoichiometry of each protein–protein interaction is defined as the ratio of transcript abundances (in FPKM units) between the two interaction partners. In case of an interaction between a homeologous gene pair and a single-copy gene, we calculated both the total stoichiometry, which combines the total abundances across homeologous genes, and the subgenome-specific stoichiometry, which considers only the homeologs that belong to the same subgenome. For example, given an interaction between a homeologous gene pair whose subgenome A and B homeologs are expressed at 10 FPKM and 20 FPKM, respectively, and a single-copy gene located on subgenome A whose expression level is 5 FPKM, then the total stoichiometry would be 30-to-5, or 6, and the subgenome-specific stoichiometry would be 10-to-5, or 2, respectively. To test whether protein–protein interaction stoichiometry was conserved following genome hybridization, we computed the stoichiometry ratio between the observed stoichiometry of a hybrid species and that of its non-hybrid relative.

**Gene functional annotation and enrichment analyses**. Each gene ortholog group was annotated with Pfam via a web service[48], and gene ontology (GO) via BLASTP[49] against *S. cerevisiae* gene functional annotation (downloaded from UniprotKB). *E*-value cutoffs were set at 1e−5 in all cases. Amino acid sequences of 7797 *T. asahii* orthologs were used as queries, and 5122 of them were annotated. For each gene group of interest, the enrichment of functional annotations was evaluated using a series of hypergeometric tests on each Pfam and GO term, followed by Bonferroni corrections of the resulting *p*-values. The significance thresholds for the adjusted *p*-value were set at 0.05 in all cases. We also removed annotation terms that correspond to fewer than 10 genes or more than 500 genes as

they may be too specific or too broad, respectively. For the in-depth characterization of 10 two-copy homeologs with significant divergent expression level in both *T. coremiiforme* and *T. ovoides* that were annotated as transmembrane transports (Supplementary Table 4), their amino acid sequences were searched against non-redundant (nr) protein database using BLASTP to obtain more details on their functions. For the analysis of *T. ovoides* homeolog groups that are more transcriptionally active on subgenome B, we applied either a 1-fold or a 2-fold threshold to the subgenome B-to-subgenome A expression ratio to select such homeolog groups and performed separate functional enrichment analyses. In both cases, there was no significant enrichment.

**Transcription factor annotation and analysis**. Known transcription factors in *S. cerevisiae* were retrieved from YEASTRACT database[50] and mapped to *Trichosporon* genes using gene ortholog mapping described in the above sections. This revealed that orthologs of 18 transcription factors are present in *Trichosporon* species (Supplementary Table 1). We then searched for the binding sites of these transcription factors in the 1 kb upstream region from the start codon of all two-copy homeolog pairs (4686 in *T. coremiiforme* and 3903 in *T. ovoides*) using the matrix-scan function of RSAT[51]. Because our RNA-seq data indicated that the distance between transcription start site and start codon is generally short, the choice of 1 kb should sufficiently cover the majority of upstream binding sites. Background nucleotide frequency model for RSAT was estimated from the input sequences using a second-order Markov model. *p*-Value cutoff was set at 1e−4. Binding motifs were selected from the 2018 JASPAR core nonredundant fungi dataset curated by RSAT. Both strands of input sequences were subjected to the searches.

**Statistics and reproducibility**. Transcriptome profiles were highly reproducible across two biological replicates for all samples. Part of the evaluation of reproducibility can be found in Supplementary Fig. 1. Also, our analyses found consistent results and conclusions for transcriptome data from both log and stationary growth phases. Statistical analyses were performed on R or MATLAB. Paired tests were used for comparing homeologous genes and unpaired tests were used otherwise.

**Reporting summary**. Further information on research design is available in the Nature Research Reporting Summary linked to this article.

## Code availability

All major data processing and analysis steps in this study were performed on publicly available software.

## Data availability

Raw RNA sequencing data are available in the Sequence Read Archive under accession number DRA007586. Processed gene expression levels, estimated evolutionary rates, and inferred protein-protein interactions for Figs. 2, 3, and Supplementary Fig. 6 are provided as Supplementary Data 1–5.

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

## Acknowledgements

The authors thank Yutaka Suzuki who performed RNA-sequencing. This work was supported by the Japan Society for the Promotion of Science (grant numbers 14F04382, 16H06154, 16H06279, and 17H05834), the Ministry of Education, Culture, Sports, Science and Technology in Japan (Research Grant to RIKEN Center for Life Science Technologies, Division of Genomic Technologies), the Japan Science and Technology Agency (CREST), and the Canon Foundation.

## Author contribution

S.S. performed data analyses. M.T., R.M. and M.O. conducted laboratory experiments and produced sequence data. S.S. and W.I. wrote the manuscript. W.I. directed and supervised the research.

## Additional information

**Competing financial interests:** The authors declare no competing financial or non-financial interests.

