## [Peer Review File · Communications Biology]

Reviewers' comments:

Reviewer #1 (Remarks to the Author):

The manuscript "Separate evolutions of genome and transcriptome in recently hybridized fungal allopolyploids" by Professor Iwasaki and co-authors investigated the contribution of Regulatory network and protein-protein interaction in two genomes went through recent hybridization. They employed RNAseq (two conditions: log-phase versus stationary-phase) to monitor the transcription regulation.

As authors pointed out, understanding evolutionary consequences after genome duplication or genome hybridization is extremely interesting. Unfortunately, some of the sections are too descriptive and lacking scientific insights and at the same time, some important details are not provided and some important analyses are not performed, which made it very hard to evaluate the validity of some of the conclusions.

The paper is based on genome division into subgenomic regions A and B, then further divide genes into single copy versus two copy genes. There are so many questions:

- How big is A how big is B for both genomes?
- How many genes in A, how many genes in B are single copy genes?
- Gene lose after hybridization will impact the expressed. Since one genome lost more genes than the other, can we compare them equally?
- What contribute to the increased correlation scores in both subgenomic regions?
- Do they share transcription factor binding sites?
- Do they retain the same transcription factors?

The paper in its current form is not easy to follow. Fundamentally, the paper lacks mechanistic understanding nor functional significance of their observations.

Reviewer #2 (Remarks to the Author):

1. Brief summary of the manuscript

Genome and transcriptome "shock" are important consequences of polyploidization and have been more extensively studied in plant species where polyploidization played an important role on evolution and diversification of plants and domesticated crops.

In fungi, polyploidization has been more intensively studied in the context of the whole genome duplication (WGD) in the Saccharomycotina and in biotechnological important species such as those belonging to the *Saccharomyces* genus. Another well studied allopolyploid is the ascomycete *Epichloë festucae* in which the evolutionary steps at the transcriptome level were rather conservative as most genes retained the parental expression levels while divergent expression between the two homeologs in the parents was apparently lost in the allopolyploid, therefore contributing to a balanced expression.

In the present paper, Iwasaki and colleagues study the impact of polyploidization in the transcriptome of two basidiomycete allopolyploids which resulted from two recent and independent inter-species hybridization events. They found overall balanced gene expression in concordance with previous studies. Also, in the hybrid that was originated from the two more distantly related parents (*T. ovoides*), the authors found a more concerted expression between the two subgenomes when compared to one of the subgenome and its respective parental reference, which was interesting and indicates reconciliation of the parental regulatory networks in the hybrid.

2. Overall impression of the work

Overall, the methodologies employed were adequate for the questions the authors propose to address. The lack of information regarding the two reference parental species for each of the hybrids (only one reference was used) limits the set of questions one can ask to unveil how transcription of each of the subgenomes evolved in comparison to the parental strains. This turns it more difficult to properly correlate the differences/similarities observed in the two subgenomes of the hybrids. However, the overall conclusions are rather supported by the data provided.

A previous work was already published by the authors describing the impact of hybridization at the genome level. The two studies complement each other; however, it would be easier to follow the present paper if both genomic and transcriptomic data were published together. In the discussion section the authors made an effort to correlate their findings in the present work with the findings on the previous paper.

Minor comments:

1. Loss of gene expression from one homeolog in the allopolyploid can also affect a broader genomic region than the gene itself. Did you find evidence for physical link in the reference genome for the genes that were found to be divergently expressed?
2. Any cases of extreme differential expression where only one of the homeologs is expressed? Any category of genes stands out?
3. Given the low divergence between the two subgenomes from *T. coremiiforme*, was it possible to clearly distinguish between in-paralogs and out-paralogs (homeologs)? I believe these cases were removed from the analysis for both hybrids as described in the materials and methods, anyway I am just curious whether you find cases of gene duplication and if so, were they more frequent in one of the subgenomes.
4. Line 83. I am not sure 17% aminoacid divergence is considered a small divergence in this context.
5. Line 112. An evaluation of the correlations of transcript expressions between homeolog of each hybrid species and other non-hybrid *Trichosporon* species was performed in order to identify the best reference for subsequent comparative transcriptomic analyses. I believe that for the purpose of understanding the evolution of transcriptome in hybrid species the best references would be the parental species (which ended up being the case)?
6. Line 151-156. This is one such conclusion that would greatly benefit from information about the missing parental reference.
7. Line 163 "transcriptional activities in *T. ovoides* are significantly higher on subgenome B". Any ideas why this is happening? Any genes/families of genes contributing to this result?
8. Line 317. Maybe there is not enough data to conclude that the enrichment in transmembrane transporters among genes that showed higher expression divergence is a mechanism of adaptation to an enlarged genome. Also, I suggest the addition of more detailed data about the genes used in the GO analysis. The authors state that a subsequent BLAST analysis was performed for these genes in order to clarify their function (line 511); I believe this information should be available for the readers.
9. Line 510 annotated as transmembrane transports (Supplementary Table 2) – Is this referring to Supplementary Table 1?
10. Supplemental Table 1- change to supplementary table 1

Responses to reviewers' comments:

Overall response:

The authors would like to thank both reviewers for their comments. Many are critical suggestions that helped further improve the analyses and the writing of the manuscript. We have addressed all comments to the best of our ability and the point-by-point responses are listed next to each comment below.

Reviewer #1

The manuscript “Separate evolutions of genome and transcriptome in recently hybridized fungal allopolyploids” by Professor Iwasaki and co-authors investigated the contribution of Regulatory network and protein-protein interaction in two genomes went through recent hybridization. They employed RNASeq (two conditions: log-phase versus stationary-phase) to monitor the transcription regulation.

As authors pointed out, understanding evolutionary consequences after genome duplication or genome hybridization is extremely interesting. Unfortunately, some of the sections are too descriptive and lacking scientific insights and at the same time, some important details are not provided, and some important analyses are not performed, which made it very hard to evaluate the validity of some of the conclusions.

The paper is based on genome division into subgenomic regions A and B, then further divide genes into single copy versus two copy genes. There are so many questions:

1. How big is A how big is B for both genomes?

Response: We have included the sizes of subgenomes A and B for both hybrids to the manuscript. *T. coremiiforme*'s subgenomes A and B contain 5,130 and 5,083 genes, respectively. *T. ovoides*'s subgenomes A and B contain 4,862 and 4,386 genes, respectively. This information has also been added to Figure 1E. [Lines 104-106]

2. How many genes in A, how many genes in B are single copy genes?

Response: We have included the number of single-copy gene to the manuscript. *T. coremiiforme*'s subgenomes A and B contain 444 and 397 single-copy genes, respectively. *T. ovoides*'s subgenomes A and B contain 959 and 483 genes, respectively. These numbers have also been added to Figure 1E. [Lines 106-109]

3. Gene lose after hybridization will impact the expressed. Since one genome lost more genes than the other, can we compare them equally?

Response: When we analyzed the evolutionary rate (dN/dS) divergence and expression divergence, the ratios were normalized by their median values to reduce intrinsic bias in evolutionary rate or transcriptional activity toward particular subgenome. This consideration can be seen in Figure 2A and 2C where the thresholds for calling significant

divergence are not symmetric around the x- and y-axes. To emphasize this important issue more clearly, we have added this rationale for normalizing the divergence ratios to the manuscript [Lines 180-182] as well as a brief statement to the discussion. [Lines 293-295]

4. What contribute to the increased correlation scores in both subgenomic regions?

Response: The increased correlation in transcriptional activity between subgenomes can be facilitated by reconciliation of both cis- and trans-regulatory elements. For example, gene conversion may occur in the upstream regulatory regions of homeologous genes and homogenize transcription factor binding sites. Analysis of transcription factor binding sites (as suggested in comment #5 below) also showed the upstream regions of homeologous genes tend to contain the binding sites of the same transcription factors [Lines 144-147]. This would allow homeologs to be co-regulated by the same transcription factors. Further, in depth investigation into the specific mechanisms would require functional genomics experiments and/or transcript expression data of these species under diverse environmental conditions, which we think are beyond the scope of current study.

5. Do they share transcription factor binding sites?

Response: We extracted the upstream regions of two-copy homeolog pairs in both hybrid species and searched for the binding sites of 18 transcription factors (detailed in our response to comment #6 below) and found that there is a significant extent of binding site sharing across homeologous subgenomes. Details are provided in Supplementary Table 2. We have also added a paragraph in the result section [Lines 134-147] and the method section [Lines 477-490] to discuss these transcription factor analyses.

6. Do they retain the same transcription factors?

Response: We were able to map 18 known transcription factors in *S. cerevisiae* (retrieved from YEASTRACT database) to their orthologs in *Trichosporon* species. This reveals that *T. coremiiforme* still retains both copies of TF (14 out of 18 detected TFs remain as two-copy) while *T. ovoides* has lost half of its TFs (8 out of 18 detected TFs are now single-copy). Among these losses of seven TF genes, six occurred on subgenome B and only one occurred on subgenome A. We have included this finding as a supporting evidence that subgenome A of *T. ovoides* is the dominant subgenome [Line 157].

7. The paper in its current form is not easy to follow. Fundamentally, the paper lacks mechanistic understanding nor functional significance of their observations.

Response: Mainly due to the lack of gene functional annotation data for *Trichosporon* species, we were not able to obtain much insight from functional enrichment analyses of important gene groups. As *Trichosporon* species belong to a different division than other well-characterized, our attempts to infer functional annotation from known fungal species using sequence similarity and protein domain prediction could be not effective. Nonetheless, we believed that our original analyses, together with new results that were obtained through suggestions from both reviewers, have clearly illustrated systematic differences in evolutionary consequences of genome hybridization in closely related hybrids. Even though

there is no definite explanation of molecular mechanisms that underlie these observations, this study should still contribute to our better understanding of genome hybridization and evolution.

Reviewer #2

Brief summary of the manuscript

Genome and transcriptome “shock” are important consequences of polyploidization and have been more extensively studied in plant species where polyploidization played an important role on evolution and diversification of plants and domesticated crops. In fungi, polyploidization has been more intensively studied in the context of the whole genome duplication (WGD) in the Saccharomycotina and in biotechnological important species such as those belonging to the *Saccharomyces* genus. Another well studied allopolyploid is the ascomycete *Epichloe festucae* in which the evolutionary steps at the transcriptome level were rather conservative as most genes retained the parental expression levels while divergent expression between the two homeologs in the parents was apparently lost in the allopolyploid, therefore contributing to a balanced expression.

In the present paper, Iwasaki and colleagues study the impact of polyploidization in the transcriptome of two basidiomycete allopolyploids which resulted from two recent and independent inter-species hybridization events. They found overall balanced gene expression in concordance with previous studies. Also, in the hybrid that was originated from the two more distantly related parents (*T. ovoides*), the authors found a more concerted expression between the two subgenomes when compared to one of the subgenome and its respective parental reference, which was interesting and indicates reconciliation of the parental regulatory networks in the hybrid.

Overall impression of the work

Overall, the methodologies employed were adequate for the questions the authors propose to address. The lack of information regarding the two reference parental species for each of the hybrids (only one reference was used) limits the set of questions one can ask to unveil how transcription of each of the subgenomes evolved in comparison to the parental strains. This turns it more difficult to properly correlate the differences/similarities observed in the two subgenomes of the hybrids. However, the overall conclusions are rather supported by the data provided.

A previous work was already published by the authors describing the impact of hybridization at the genome level. The two studies complement each other; however, it would be easier to follow the present paper if both genomic and transcriptomic data were published together. In the discussion section the authors made an effort to correlate their findings in the present work with the findings on the previous paper.

Minor comments:

1. Loss of gene expression from one homeolog in the allopolyploid can also affect a broader genomic region than the gene itself. Did you find evidence for physical link in the reference genome for the genes that were found to be divergently expressed?

Response: We visualized the trend of transcript expression divergence along *T. coremiiforme*'s and *T. ovoides*'s genomes and found a 370kb region on scaffold 7 of *T. coremiiforme* where transcription activities are consistently suppressed. There are 138 genes located in this region, but gene functional analysis did not show any significant enrichment at $p\text{-value} \leq 0.05$. For *T. ovoides*, probably due to intrinsic bias in transcriptional activity toward subgenome B, no distinctive region could be observed. We have added this finding to the Result section [Lines 170-174] and provided the details in an additional supplementary figure (new Supplementary Figure 3).

2. Any cases of extreme differential expression where only one of the homeologs is expressed? Any category of genes stands out?

Response: We have examined the cases where the gene sequences of both homeolog copies could be found in the genome but the transcript of only one homeolog copy was detected in the transcriptome (called "singly-expressed homeolog groups"). There are 32 such singly-expressed homeolog groups in *T. coremiiforme* and 25 in *T. ovoides*. Then, we compared the expression levels of the singly-expressed homeolog groups with those of other genes and found that the expression levels of almost all singly-expressed homeolog groups are well below the 25th percentile and lower than 1 FPKM (Figure 1 below). This suggested that the reason why we could not detect the expression of one homeolog copy was more likely due to the sensitivity of RNA-sequencing than due to extreme expression divergence. Also, there was no significant enrichment of any gene function for these homeolog groups (but this could be due to small gene set sizes and low coverage of functional annotation in *Trichosporon* as well). We have added a brief mentioning of this result to the manuscript. [Lines 166-170]

Figure 1: Comparison of expression levels between singly-expressed homeolog groups against the genome averages. Red bars indicate the median. Black boxes indicate the 25th-75th percentile range. Black whiskers indicate the 5th-95th percentile range. Black circles represent individual outliers.

3. Given the low divergence between the two subgenomes from *T. coremiiforme*, was it possible to clearly distinguish between in-paralogs and out-paralogs (homeologs)? I believe these cases were removed from the analysis for both hybrids as described in the materials and methods, anyway I am just curious whether you find cases of gene duplication and if so, were they more frequent in one of the subgenomes.

Response: The reviewer's understanding is correct that these cases where confusion can occur between in-paralogs and out-paralogs have been removed from the analysis. We specifically ignored ortholog groups where there are multiple genes in a non-hybrid species and ortholog groups where there are more than two genes in a hybrid species. These cases constitute only 348 out of 7,857 ortholog groups identified in our analysis and their exclusion should not impact the study's conclusions. We have clarified our rationale in the method section. [Lines 330-333]

To answer the reviewer's question about gene duplication frequency in each subgenome (of *T. coremiiforme*), we examined 43 ortholog groups where there are one gene copy in *T. asahii* and 3 gene copies in *T. coremiiforme*, and checked whether the extra gene copy in *T. coremiiforme* is located on subgenome A or B. Synteny structure and the subgenome assignment of nearby genes were used to assign these gene copies. This showed that 26 of them are located on subgenome B, 13 on subgenome A, and 4 could not be assigned. Overall, there are only small number of cases of gene duplications and subgenome B of *T. coremiiforme* seems to contain more of them. These results were not added to the manuscript.

4. Line 83. I am not sure 17% amino acid divergence is considered a small divergence in this context.

Response: We agree with the reviewer that the level of amino acid divergences between *Trichosporon* species observed here are not considered small in this context and we have removed the word "small" from this sentence. [Line 78]

5. Line 112. An evaluation of the correlations of transcript expressions between homeolog of each hybrid species and other non-hybrid *Trichosporon* species was performed in order to identify the best reference for subsequent comparative transcriptomic analyses. I believe that for the purpose of understanding the evolution of transcriptome in hybrid species the best references would be the parental species (which ended up being the case)?

Response: We agree that the best references would be the parental species and that our explanation can cause unnecessary confusion. We have revised this sentence. [Line 110]

6. Line 151-156. This is one such conclusion that would greatly benefit from information about the missing parental reference.

Response: To better convey the significance of this conclusion, we added reference to Figure 1A which showed the large phylogenetic distance that separate the two subgenomes of *T. ovoides* from each other and from other relatives. We also added a statement, “which resulted in high gene loss rates predominantly from one of its subgenomes (*I9*)”, to highlight that the two subgenomes of *T. ovoides* are clearly evolutionarily diverged. [Lines 129-130]

7. Line 163 “transcriptional activities in *T. ovoides* are significantly higher on subgenome B”. Any ideas why this is happening? Any genes/families of genes contributing to this result?

Response: We have performed functional enrichment analyses for *T. ovoides*'s two-copy homeolog groups where the subgenome B gene copy has higher expression level. We tried two thresholds for selecting such homeolog groups: 1-fold and 2-fold. At 1-fold threshold, there were 2,470 homeolog groups with higher expression on subgenome B. At 2-fold threshold, there were 1,258 such groups. In both cases, there was no significant enrichment of any gene function. We have added a brief statement on this issue to the Result section [Lines 163-165] and added detailed explanations to the Method section [Lines 472-476].

8. Line 317. Maybe there is not enough data to conclude that the enrichment in transmembrane transporters among genes that showed higher expression divergence is a mechanism of adaptation to an enlarged genome. Also, I suggest the addition of more detailed data about the genes used in the GO analysis. The authors state that a subsequent BLAST analysis was performed for these genes in order to clarify their function (line 511); I believe this information should be available for the readers.

Response: We do not have conclusive evidence for the mechanism behind divergent expressions of transmembrane proteins. To further clarify that this statement about transmembrane transporters is not a conclusion, but rather a possibility, we have revised the statement in question [Lines 271-275]. The data on detailed functions of identified transmembrane transporters is now included in Supplementary Table 4.

9. Line 510 annotated as transmembrane transports (Supplementary Table 2) – Is this referring to Supplementary Table 1?

Response: We are sorry for the oversight. Somehow the Supplementary Table 2 was missing from the Supplementary Material file. We have added this table back into the new version (now Supplementary Table 4).

10. Supplemental Table 1- change to supplementary table 1

Response: We have changed all “Supplemental” to “Supplementary”.

REVIEWERS' COMMENTS:

Reviewer #1 (Remarks to the Author):

The authors have carefully address all my comments and accepted all my suggestions. No further questions.

Reviewer #2 (Remarks to the Author):

All the questions were addressed by the authors, some of which involved new analyses that helped to clarify some of the issues that were raised. The text itself did not improve much and despite the fact that some modifications were made to include new details/results, I still feel that some of the result sections are somewhat confusing. Personally, protein-protein interaction results section was particularly difficult to follow.